# Evaluating the Impact of Crop Layout Changes on N and P Nutrient Balance: A Case Study in the West Liaohe River Basin, China

**Zijuan Zhao** [1], **Beilei Fan** [1] **and Dong Liu** [2,*]

1   Agricultural Information Institute of CAAS, Beijing 100081, China;
    zhaozijuanyouxiang@163.com (Z.Z.); fanbeilei@caas.cn (B.F.)
2   College of Resources and Environment, University of Chinese Academy of Sciences, Beijing 100049, China
*   Correspondence: lldking@ucas.ac.cn; Tel.: +86-10-69672953

**Abstract:** Regional crop layout has changed significantly due to climate, policy, and other factors, which has impacted farmland nutrient balance. Here, we evaluated the impact of crop layout changes on N and P nutrient balance in the West Liaohe River Basin from 2000–2015. The study area has long been in a N and P surplus state. The unit N surplus exhibited a downward trend and the unit P surplus showed an increasing trend. Significant correlations existed between planting areas and nutrient surplus. The N and P surplus layout was mainly concentrated in the West Liaohe River lower reaches basin. The planting area of wheat must be reduced and the areas of maize and soybean must be controlled to adjust the N and P balance and reduce the environmental pollution risk. Chemical fertilizer and seed inputs are the main sources of N input. Furthermore, combining farming and pastoral farming is conducive to improving N and P use efficiency. Manure can be absorbed by farmland, the ratio of organic and chemical fertilizers can be reasonably set, and chemical fertilizer application can be reduced.

**Keywords:** crop layout; N and P balance; chemical fertilizer; N and P surplus; N and P use efficiency

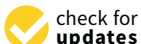



## 1. Introduction

Protecting cropland and improving cropland quality are the basis for global food security and sustainable agricultural development [1,2]. To promote crop growth and increase crop yields, the use of chemical fertilizers has become the main method for stable agricultural production [3]. The global population is expected to increase from 7.8 billion currently to 9.7 billion by 2050, with further growth to 10.9 billion by the end of the century; global demand of crop production is predicted to increase accordingly [4–6]. Globally, prospects for increasing the gross cultivated area are limited [7]. Since the 1980s, agricultural production in China has increasingly depended on the use of chemical fertilizers to meet food demands [8–10]. In 2014, total chemical fertilizer consumption in China accounted for 31% of the global total [11]. The use of large amounts of chemical fertilizers has greatly increased crop yields. However, in the case of low N and P use efficiency, excessive fertilization not only wastes nutrients but also causes serious environmental problems such as groundwater pollution, eutrophication of surface waters, decline in biodiversity, water quality degradation, soil acidification, and an increase in greenhouse gas emissions [12]. Farmland nutrient balance is an important indicator of the degree of farmland pollution. Numerous studies have shown that farmland in most parts of China has been in an N and P nutrient surplus in recent years [13,14]. Since 1980, N and P nutrient inputs have been greater than the output in China. From 1980–2010, the surplus of N and P nutrients increased significantly, with the largest increase in Northeast China followed by North and Northwest China [15]. Numerous scholars have also studied the farmland nutrient balance worldwide. Lederer et al. [16] found that like many other countries in sub-Saharan Africa

(SSA), Uganda faces a remarkable soil nutrient deficit in farmland soils. Tran T.M. et al. [17] assessed the crop nutrient requirements in Dong Nai and Bac Ninh provinces of Vietnam, estimated the farmland nutrient balance in the two provinces, and spatially expressed it based on GIS.

The soil surface nutrient balance is mainly calculated by the difference between nutrient input and output in agricultural systems [18–20]. In agricultural systems worldwide, farm gate, soil surface, and soil systems are the dominant methods of budgeting nutrients [21,22]. Previous studies have suffered from difficulties in obtaining crop cultivation statistics and limited spatial scale expression of farmland nutrient balance [23]. Most studies have been conducted through field experiments to study the effects of crop layout on farmland nutrient balance. There have been fewer and more problematic studies at the macro level by domestic and foreign scholars [24–26]. As the basic administrative unit, the county is the smallest unit of agricultural policy implementation. Therefore, many scholars in China and abroad have taken the county as the smallest research unit to conduct research on farmland nutrient balance [27]. However, taking the county as the unit can only reflect the overall farmland nutrient balance in the county and cannot determine the spatial distribution changes of farmland nutrients in the county. Remote sensing crop monitoring studies have been conducted to obtain large-scale crop information in a rapid, accurate, and timely manner. With the increasing abundance of remote sensing data sources, research on remote sensing technology for crop identification has developed from single to multiple crop identification [28]. Numerous studies in China and abroad have shown that the use of MODIS data combined with phenological information can effectively extract crop planting layout in large regions [29–31]. Using GIS builds a database on agricultural nutrients, long-term monitoring of agricultural nutrients over time, a comparison of these data over several periods, and a large-scale analysis of the spatial coverage and variation in space [32]. The nutrient balance of farmland is expressed at the pixel scale and can assist relevant departments to collect timely information on the nutrient balance of farmland and scientifically assess the nutrient status of regional farmland and environmental risks.

The Western Liaohe Plain is located in the World Golden Maize Belt and is an arid and semi-arid region. In recent years, owing to the influences of climate and policies, the crop layout in this area has changed significantly. Changes in crop planting layout inevitably have an impact on the farmland nutrient balance. If N and P inputs exceed the needs of crop production, excess nutrients accumulate in the soil and the remaining nutrients will be discharged into water bodies through gas emissions, leaching, and runoff, causing environmental pollution [33–35]. Soybean, wheat, and maize are the main food crops in the West Liaohe River Basin. In this study, we combined the advantages of remote sensing to interpret crop layout with MODIS data, constructed a model of farmland nutrient balance, and built a nutrient balance database based on GIS. This study is of great significance for improving the utilization rate of fertilizers, reducing environmental pollution, developing and utilizing regional arable land, and efficiently utilizing agricultural resources in arid and semi-arid regions. This research has significant implications for food security, sustainable crop production, and other sustainable use of resources.

## 2. Materials and Methods

### 2.1. Study Area

The West Liaohe River Basin is located at the junction of the eastern section of the farming–pastoral ecotone in northern China [36]. The total area of the basin (41°05′–45°12′ N, 116°36′–124°34′ E) is $1.36 \times 10^5$ km$^2$ and covers 23 counties in the Inner Mongolia Autonomous Region, Hebei, Liaoning, and Jilin provinces (Figure 1). The mountainous area of the basin is approximately $8.4 \times 10^4$ km$^2$. Most of the West Liaohe River Basin water system is located in the northeastern part of the Inner Mongolia Autonomous Region. The 1st level tributaries of the West Liaohe River include the Laoha, Xilamulun, Jiaolai, and Xinkai Rivers and are divided into three sub-basins: Xilamulun and Laoha River Basins (upper reaches), Wulijimu River Basin (middle reaches), and West Liaohe River lower

reaches basin (below Sujiabao) (lower reaches). The overall topography of the Liaohe River Plain exhibits a gently inclined trend from west to east, with the lowest in the eastern region and is undulating with an elevation of approximately 82 m. The West Liaohe River Basin is located in a continental monsoon climate zone. It is restricted by the Mongolian plateau airflow, with low precipitation and large seasonal changes [37]. Geographical and climatic conditions are suitable for maize cultivation. The main crops in this region are maize, soybean, and wheat.

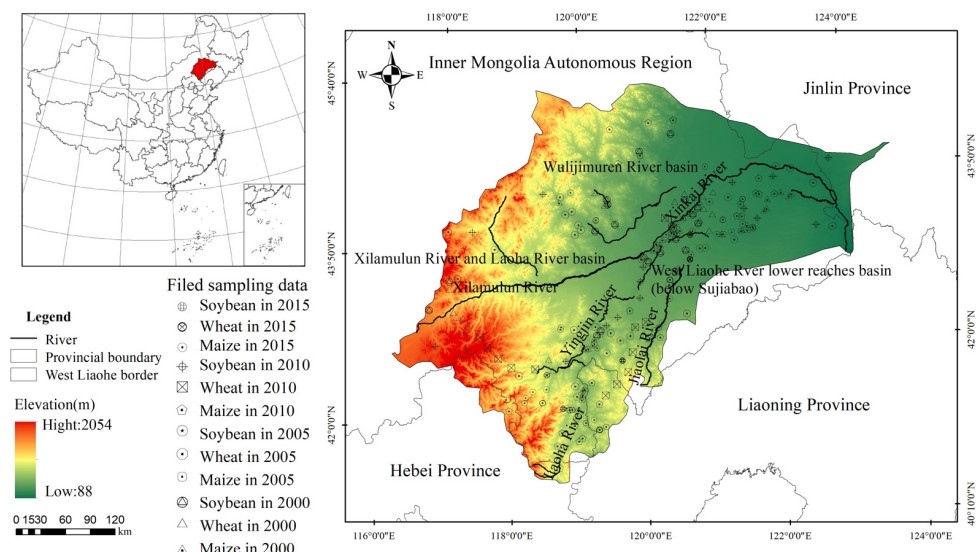

**Figure 1.** Location of the study area.

### 2.2. Data Sources

The datasets used to evaluate the impact of crop layout changes on N and P nutrient balance were obtained from the National Aeronautics and Space Administration (http://modis.gsfc.nasa.gov/ (accessed on 5 July 2015)), United States Geological Survey (https://earthexplorer.usgs.gov/ (accessed on 7 July 2015)), the China Agricultural Products Cost Benefit Compilation of Information, the China County Statistical Yearbook, and field survey data. Four types of data sets were collected, including: remote sensing image data, GPS sampling point data, statistical data of crop planting area, and farmland nutrient balance data. In July 2010 and June 2015, we obtained 467 GPS sampling point data of crops used as training and verification data (Figure 1). It is worth noting that crop GPS sampling point data for 2000 and 2005 were determined according to the memories of farmers. With 2010 as the base period, if an area had the same crop planted for five consecutive years, it was used as a sampling point for 2005. If an area had the same crop planted for 10 continuous years, it was a sampling point for 2000. We interviewed farmers from the agricultural, pastoral, semi-agricultural, and semi-pastoral areas in the West Liaohe River Basin and collected farmland nutrient data including fertilizer types and amounts, crop types and yields, livestock types and amounts. Similarly, farmland nutrient data from 2000 and 2005 were also determined according to farmers' memories. In this study, the three agricultural regions were divided according to the distribution of the main tributaries, which enabled evaluation of the impact of crop layout changes on N and P nutrient balance. The three regions considered in this analysis were the Xilamulun and Laoha River Basins (upper reaches), Wulijimu River Basin (middle reaches), and West Liaohe River lower reaches basin (below Sujiabao) (lower reaches). Table 1 summarizes the datasets used in the study.

**Table 1.** Datasets used in this study.

| Data | Time Period | Scale | Source |
|---|---|---|---|
| MOD09Q1 | 2000, 2005, 2010, 2015 | 250 m | http://modis.gsfc.nasa.gov/ (accessed on 5 July 2015) |
| Landsat 8 OLI | 2000, 2005, 2010, 2015 | 30 m | https://earthexplorer.usgs.gov/ (accessed on 7 July 2015) |
| Statistical data of crop planting area | 2000, 2005, 2010, 2015 | Yearly | China County Statistical Yearbook |
| GPS sampling point data of crops | 2000, 2005, 2010, 2015 | - | Field survey in July 2010 and June 2015 |
| Farmland nutrient data | 2000, 2005, 2010, 2015 | - | China Agricultural Products Cost Benefit Compilation of information; field survey in July 2010 and June 2015 |

*2.3. Methods*

Figure 2 illustrates the framework of mapping cropland and crop types and the N and P nutrient balance model. In the first step, the cropland boundary of the study area was extracted based on Landsat 8 OLI [38]. In the second step, crop types were extracted and a crop layout database was constructed based on MOD09Q1 data within the cropland boundary. In the third step, based on the crop layout database, the parameters of N and P input–output were entered into the database, and the N and P farmland nutrient balance of the study area was calculated in the GIS environment. The innovation of this method is that it accurately reflects the regional N and P nutrient balance and potential pollution risk based on the pixel scale.

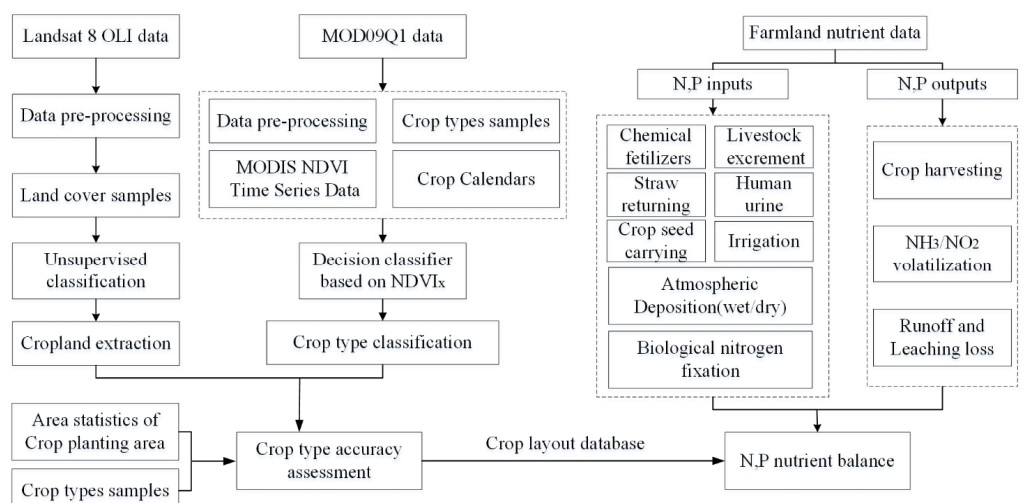

**Figure 2.** Framework of extracting croplands, crop types, and N and P balance model.

2.3.1. Approach to Mapping Crop Type

MOD09Q1 is a land surface reflectance L3 product, and these data are 8 d time series data at a 250 m spatial resolution. MOD09Q1 data has a high time resolution and rich spectral bands, which can be used to detect the seasonal dynamics of normalized difference vegetation index (NDVI) in different crops [39,40]. These data had previously undergone atmospheric correction [41]. Tile h26v04 and h27v04 were masked to cover the entire study area and were projected to the Albers conical equal area projection using the Modis Reprojection Tool [42], and the geographic coordinate system was WGS-84. The converted data were the batch closest with the study area as the cropping frame. Calculation of the NDVI uses the red (620–670 nm) and near-infrared (841–875 nm) bands based on the following equation [43,44]:

$$\text{NDVI} = \frac{NIR - RED}{NIR + RED},\tag{1}$$

where *RED* is the near-infrared reflectance and *NIR* is the reflectance corresponding to red.

We located the farm records of crops in the West Liaohe River Basin in the database provided by the Ministry of Agriculture and Rural Affairs of the Republic of China (http://www.zzys.moa.gov.cn/ accessed on 10 July 2015). In spring, maize is usually

sown in early May, emerges in mid-to-late May, grows rapidly from early June to late July, and is finally harvested in mid-September. Wheat is sown in mid-April, emerges in mid-late May, develops tillers in June, elongates and milks from early July to late July, and matures in mid-August. Soybean is normally sown and emerges in late May, flowering occurs from late June to mid-July, podding in mid-August, and harvest is conducted in early September. The typical crop season length of all crops is approximately 100–120 d (Figure 2). Every crop type has its own characteristics in the sowing, emergence, elongation, tasseling, and harvest stages [45]. Annual NDVI time-series data can clearly reflect the whole process of crop growth through the dynamic process of "elevation-reaching peak-lowering" [46]. Figure 3 illustrates the NDVI temporal profiles based on the three crop types. It is worth noting that a single cropping system has only one peak stage. Different crops have different phenological rhythms, which are reflected in different patterns of NDVI time series curves, and such patterns can be used to classify crop type maps [47].

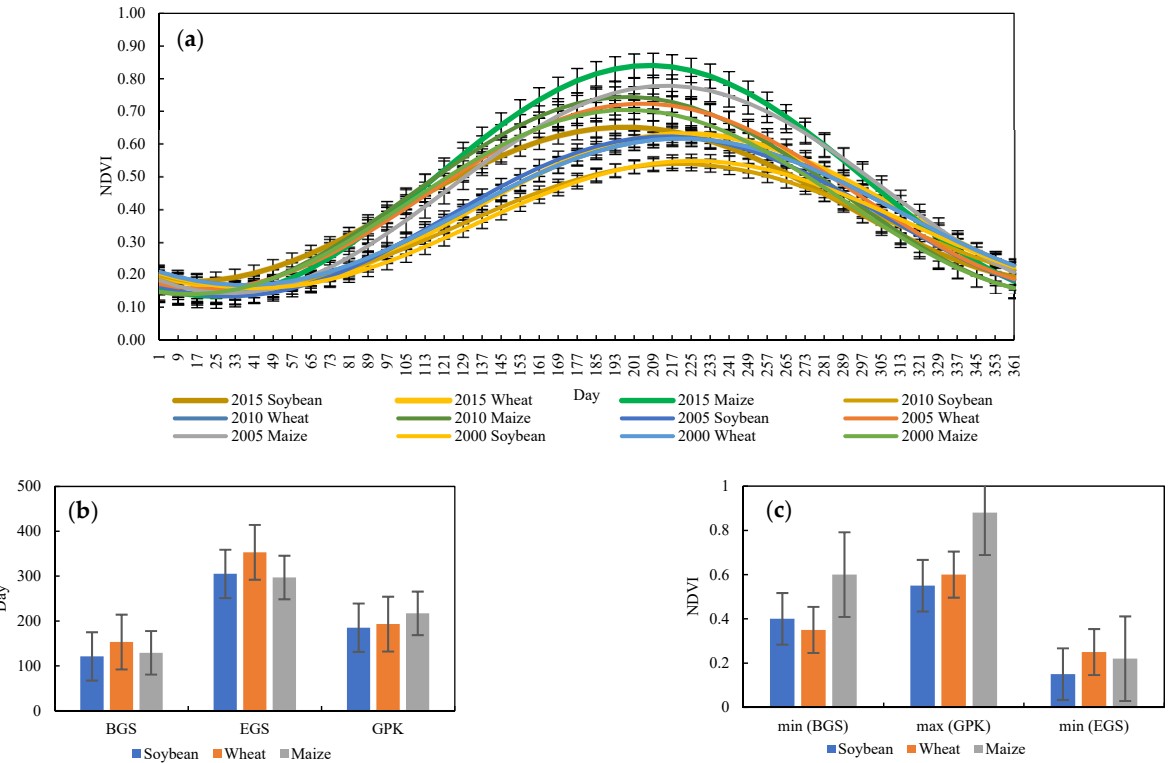

**Figure 3.** (**a**) Normalized difference vegetation index (NDVI) time series; (**b**,**c**) phenology features. BGS, start of season; EGS, end of season; GPK, peak growth period.

Figure 4 illustrates the strategy of using a decision tree based on three phenological features to classify crop types. According to the statistics shown in Figure 3, the following thresholds were determined in this study:

(1) If the BGS of the crop pixel is 129, GPK is 217, and EGS is 297, then the modified pixel is classified as maize.
(2) If the BGS of the crop pixel is 153, GPK is 193, and EGS is 353, then the modified pixel is classified as wheat.
(3) If the BGS of the crop pixel is 129, GPK is 297, and EGS is 217, the modified pixel is classified as soybean.
(4) $T_1$–$T_9$ were used to identify the NDVI thresholds of key phenological nodes (emergence, tasseling, and maturity) of crops.

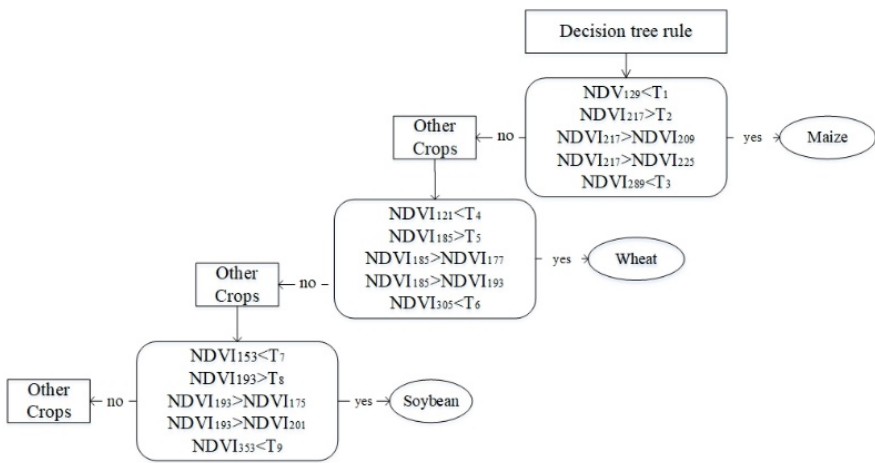

**Figure 4.** Decision tree rules of crop types classification. NDVI—normalized difference vegetation index.

Min (BGS) represents the NDVI value of the crop at the seedling stage, max (GPK) represents the NDVI value of the crop during the tasseling period, and max (EGS) represents the NDVI value of the crop at maturity (Figure 3c).

It is worth noting that the $T_1$–$T_9$ threshold values and BGE, EGS, and GPK threshold values in the formula were different in different years and regions. Differences in crop growth are caused by phenological environments in different counties. See Table 2.

**Table 2.** Crop calendar of the West Liaohe River Basin.

| Crop | April | | May | | | June | | | July | | | August | | | September | | |
|---|---|---|---|---|---|---|---|---|---|---|---|---|---|---|---|---|---|
| | 2 | 3 | 1 | 2 | 3 | 1 | 2 | 3 | 1 | 2 | 3 | 1 | 2 | 3 | 1 | 2 | 3 |
| Maize | | | Sowing | Emerging | | Emerging–Elongating | | | | Tasseling | | | Tasseling–Milking | | | Harvesting | | |
| Wheat | Sowing | | Sowing–Emerging | | | Tillering | | | Elongating–Milking | | | Harvesting | | | | | |
| Soybean | | | | | | Sowing–Emerging | | | Elongating | | | Elongating–Flowering | | | Podding–Harvesting | | |

Note: 1–3 represent the first ten days, middle ten days, and last ten days of a month, respectively.

### 2.3.2. N and P Nutrient Balance Model

The N and P nutrient balance was calculated similar to the OECD [22] and He [48] with some modifications of input and output components at the regional scale (Figure 4) using Equation (2) (see Table 3 for the detailed formula). The N and P use efficiency indicator is defined as the percentage of crop uptake to N and P input [49] and was calculated using Equation (3). The N and P surplus rate indicator is defined as the percentage of N and P balance to N and P output [50] and was calculated using Equation (4).

$$NP_{balance} = NP_{input} - NP_{output} \tag{2}$$

If the $NP_{balance}$ is 0, there is no N or P surplus in the soil. If $NP_{balance}$ is <0, it indicates potential N and P deficits in the soil. Therefore, N and P additions are needed to satisfy crop N and P requirements. An $NP_{balance} > 0$ indicates a potential N and P surplus in the soil; therefore, it may represent low N and P use efficiency or economic losses of N and P for farmers.

$$NPUE = \frac{NP_{grain\_removal} + NP_{straw\_removal}}{NP_{input}} \times 100\%. \tag{3}$$

Here, *NUE* denotes N use efficiency, *PUE* denotes P use efficiency.

$$NPSR = \frac{NP_{input} - NP_{output}}{NP_{input}} \times 100\%. \tag{4}$$

In Equation (4), *NSR* denotes the N surplus ratio, *PSR* denotes the P surplus ratio.

**Table 3.** N and P balance equations.

| N and P Nutrient Input | N and P Nutrient Output |
|---|---|
| $NP_{chemical} = Aera_{crop} \times NP_{unitfert}$ | $NP_{grain\_removal} = Aera_{crop} \times Yield_{grain} \times NP_{Grain\_i}$ |
| $NP_{live} = Aera_{crop} \times NP_{unitlive}$ | $NP_{straw\_removal} = Aera_{crop} \times Yield_{grain} \times SG_{Ratio\_i} \times (1 - SRate_{retur\_i}) \times NP_{Straw\_i}$ |
| $NP_{man} = Pop_{man} \times NPExcre_{Rate\_i} \times MRate_{retur\_i}$ | $N_{NO_2} = (N_{chemical} + N_{live}) \times NO_{2Rate\_i}$ |
| $NP_{straw} = Aera_{crop} \times Yield_{grain} \times SG_{Ratio\_i} \times SRate_{retur\_i} \times NP_{Straw\_i}$ | $N_{NH_3} = (N_{chemical} + N_{live}) \times NH_{3Rate\_i}$ |
| $NP_{seed} = Area_{crop} \times Seed_{unitarea} \times S_{Rate\_i}$ | $N_{leach} = Aera_{crop} \times NL_{Rate\_i}$ |
| $NP_{irri} = Aera_{crop} \times I_{Rate\_i}$ | $NP_{runoff} = Aera_{crop} \times NPR_{Rate\_i}$ |
| $NP_{atmo(dry+wet)} = Aera_{crop} \times DW_{Rate\_i}$ | |
| $NP_{biol} = Area_{Soybean} \times N_{Rate\_i}$ | |

Note: $Aera_{crop}$, crop type area (ha); $NP_{unitfert}$, N in chemical fertilizer (kg ha$^{-1}$), P in chemical fertilizer (kg ha$^{-1}$); $NP_{unitlive}$, N in livestock (kg ha$^{-1}$), P in livestock (kg ha$^{-1}$); $Pop_{man}$, rural population; $NPExcre_{Rate\_i}$, fecal urine production per person (kg); $MRate_{retur\_i}$, proportions returned to field (%); $Yield_{grain}$, grain yield per unit area; $SG_{Ratio\_i}$, ratio of straw to grains; $SRate_{retur\_i}$, N, in straw (%), P in straw (%); $NP_{Straw\_i}$, proportions returned to field (%); $Seed_{unitarea}$, sowing amount per unit area (kg ha$^{-1}$); $S_{Rate\_i}$, N in seeds (kg ha$^{-1}$), P in seeds (kg ha$^{-1}$); $I_{Rate\_i}$, irrigation water nutrient (kg ha$^{-1}$); $DW_{Rate\_i}$, atmospheric deposition rate (kg ha$^{-1}$); $N_{Rate\_i}$, biological nitrogen fixation (kg ha$^{-1}$); $NP_{Grain\_i}$, grain nutrient content (kg ha$^{-1}$); $NO_{2Rate\_i}$, NH$_3$ volatilization rate (%); $NH_{3Rate\_i}$, NO$_2$ volatilization rate (%); $NL_{Rate\_i}$, N leaching (kg ha$^{-1}$); and $NPR_{Rate\_i}$, N runoff (kg ha$^{-1}$), P runoff (kg ha$^{-1}$).

### 2.3.3. N and P Input–Output and Parameters

Fertilizer inputs consist of both chemical and organic fertilizer inputs. The coefficients for estimating chemical fertilizer input are shown in Table 4. There is a wide variety of application sources of organic fertilizer [51] which mainly include livestock manure, human feces, and crop straws. Table 5 lists the parameters used to estimate fertilizer inputs for livestock and poultry based on field surveys. The amount of fecal urine produced per person is 0.69 kg N and 0.46 kg P, and the return rate is 30%. Table 6 lists the rural populations in each county. The nutrient input of straw return was mainly estimated based on the nutrient content of different organic fertilizers, the corresponding input, and the ratio of straw to grain. The amount of nutrients in the straw returning to the field was calculated based on the parameters listed in Tables 7 and 8. Tables 9 and 10 list the parameters of the sowing amount per unit area and its nutrient coefficients to calculate the seed nutrient input. Based on the field survey, the values of the nutrient load from irrigation for N and P were 8.6 kg ha$^{-1}$ yr$^{-1}$ and 0.5 kg ha$^{-1}$ yr$^{-1}$, respectively. Symbiotic N fixation in uplands was estimated based on crop production and the amount of N uptake per crop production. For soybeans, the biological N fixation was 77 kg ha$^{-1}$ yr$^{-1}$. Atmospheric nutrient deposition, including wet and dry deposition, is generated from a variety of agricultural, natural, and industrial sources [52]. Studies have shown that the annual sediment load in Europe is 20–50 kg ha$^{-1}$, whereas in Asia this is much lower at 4–5 kg ha$^{-1}$. In this study, the parameters of the N, P deposition annual mean values of 9.14 and 0.58 kg P ha$^{-1}$ yr$^{-1}$ were set for the dry and wet deposition flux, respectively.

**Table 4.** Chemical fertilizer input parameters.

| Crop | Region | N Chemical Fertilizer (kg ha$^{-1}$) | | | | P Chemical Fertilizer (kg ha$^{-1}$) | | | |
|---|---|---|---|---|---|---|---|---|---|
| | | 2000 | 2005 | 2010 | 2015 | 2000 | 2005 | 2010 | 2015 |
| Soybean | 1 | 68.3 | 100.5 | 142.5 | 178.5 | 39.8 | 61.5 | 102.0 | 137.3 |
| | 2 | 59.3 | 86.3 | 123.0 | 154.5 | 30.8 | 47.3 | 78.0 | 105.8 |
| | 3 | 59.3 | 87.0 | 123.0 | 154.5 | 24.0 | 36.8 | 60.8 | 81.8 |
| Wheat | 1 | 171.0 | 250.5 | 356.3 | 446.3 | 90.0 | 139.5 | 230.3 | 310.5 |
| | 2 | 183.0 | 268.5 | 381.0 | 477.8 | 98.3 | 152.3 | 251.3 | 339.0 |
| | 3 | 157.5 | 230.3 | 327.0 | 410.3 | 75.8 | 117.0 | 192.8 | 260.3 |
| Maize | 1 | 142.5 | 208.5 | 296.3 | 372.0 | 75.0 | 116.3 | 191.3 | 258.8 |
| | 2 | 153.0 | 223.5 | 317.3 | 398.3 | 82.5 | 126.8 | 209.3 | 282.8 |
| | 3 | 131.3 | 192.0 | 273.0 | 342.0 | 63.0 | 97.5 | 160.5 | 216.8 |

Note: 1—semi-agricultural half pastoral area; 2—pastoral area; and 3—agricultural area.

**Table 5.** Livestock fertilizer input parameters.

| Crop | Region | N Livestock Fertilizer (kg ha$^{-1}$) | | | | P Livestock Fertilizer (kg ha$^{-1}$) | | | |
|---|---|---|---|---|---|---|---|---|---|
| | | 2000 | 2005 | 2010 | 2015 | 2000 | 2005 | 2010 | 2015 |
| Soybean | 1 | 16.5 | 27.8 | 24.8 | 22.5 | 10.5 | 19.5 | 18.0 | 15.8 |
| | 2 | 36.8 | 63.0 | 56.3 | 51.0 | 22.5 | 41.3 | 38.3 | 33.8 |
| | 3 | 28.5 | 48.8 | 43.5 | 39.8 | 18.8 | 33.8 | 31.5 | 27.8 |
| Wheat | 1 | 54.8 | 93.8 | 84.8 | 75.8 | 36.0 | 65.3 | 60.8 | 53.3 |
| | 2 | 84.0 | 143.3 | 129.0 | 115.5 | 51.0 | 91.5 | 85.5 | 75.0 |
| | 3 | 48.8 | 82.5 | 74.3 | 66.8 | 31.5 | 57.0 | 53.3 | 46.5 |
| Maize | 1 | 28.5 | 48.8 | 43.5 | 39.8 | 18.8 | 33.8 | 31.5 | 27.8 |
| | 2 | 46.5 | 79.5 | 71.3 | 63.8 | 27.8 | 49.5 | 46.5 | 40.5 |
| | 3 | 26.3 | 45.8 | 40.5 | 36.8 | 17.3 | 31.5 | 29.3 | 25.5 |

Note: 1—semi-agricultural half pastoral area; 2—pastoral area; and 3—agricultural area.

**Table 6.** Rural population of each county (×10,000).

| County | 2000 | 2005 | 2010 | 2015 | County | 2000 | 2005 | 2010 | 2015 |
|---|---|---|---|---|---|---|---|---|---|
| Weichang | 45.1 | 43.5 | 44.4 | 46.1 | HexingtenQi | 20.6 | 19.5 | 19.7 | 19.5 |
| Chifeng | 63.8 | 63.9 | 66.0 | 71.2 | OngniudQi | 41.8 | 41.6 | 41.3 | 43.5 |
| Ar Horqin Qi | 26.3 | 25.7 | 24.0 | 25.9 | KarqinQi | 32.9 | 33.0 | 30.6 | 30.9 |
| Aohan | 52.5 | 53.3 | 53.1 | 54.0 | Ningcheng | 52.3 | 52.9 | 52.3 | 53.9 |
| Horqin Zuoyi Houqi | 42.6 | 40.3 | 41.6 | 40.3 | Tongliao | 37.0 | 37.0 | 37.9 | 38.4 |
| HorqinZuoyiZhongqi | 31.2 | 28.7 | 27.7 | 28.3 | Kailu | 31.7 | 30.6 | 26.5 | 26.7 |
| Hure Qi | 13.2 | 12.0 | 11.1 | 14.0 | Naiman | 35.4 | 33.1 | 31.0 | 32.2 |
| Jarud Qi | 22.3 | 19.9 | 19.4 | 20.7 | BairinZuoi | 31.0 | 31.5 | 30.0 | 29.6 |
| HorqinYouyiZhongqi | 17.9 | 17.9 | 17.9 | 18.0 | Jianping | 47.0 | 46.1 | 45.7 | 45.4 |
| Pingquan | 40.1 | 33.8 | 34.2 | 42.3 | Shuangliao | 17.7 | 25.3 | 27.7 | 23.2 |
| Bairin Youqi | 13.2 | 11.9 | 12.7 | 13.1 | Tongyu | 11.3 | 22.8 | 24.9 | 24.5 |
| Linxi | 19.5 | 18.1 | 18.1 | 18.4 | | | | | |

**Table 7.** Crop yield per unit area (grain yield kg ha$^{-1}$).

| Crop | Region | 2000 | 2005 | 2010 | 2015 |
|---|---|---|---|---|---|
| Soybean | 1 | 414.0 | 326.9 | 376.3 | 360.4 |
| | 2 | 415.6 | 328.2 | 377.7 | 361.8 |
| | 3 | 384.6 | 303.7 | 349.5 | 334.8 |
| Wheat | 1 | 90.0 | 134.6 | 136.2 | 84.5 |
| | 2 | 90.0 | 134.7 | 136.3 | 84.5 |
| | 3 | 74.0 | 110.7 | 112.0 | 69.5 |
| Maize | 1 | 12.4 | 21.0 | 28.8 | 44.2 |
| | 2 | 12.4 | 21.0 | 28.8 | 44.3 |
| | 3 | 13.7 | 23.2 | 31.9 | 49.0 |

Note:1—semi-agricultural half pastoral area; 2—pastoral area; and 3—agricultural area.

**Table 8.** Straw manure input parameters.

| Crop | Ratio of Straw to Grains | N in Straw (%) | P in Straw (%) | Proportions Returned to Field (%) |
|---|---|---|---|---|
| Soybean | 1.6 | 17 | 0.20 | 17 |
| Wheat | 1.1 | 0.65 | 0.08 | 40 |
| Maize | 2 | 0.92 | 0.15 | 32 |

**Table 9.** Sowing amount per unit area (kg ha$^{-1}$).

| Crop | Region | 2000 | 2005 | 2010 | 2015 |
|---|---|---|---|---|---|
| Soybean | 1 | 2069.1 | 3094.2 | 3131.7 | 1942.7 |
| | 2 | 2060.4 | 3081.2 | 3118.6 | 1934.6 |
| | 3 | 2475.9 | 3702.5 | 3747.4 | 2324.6 |
| Wheat | 1 | 6264.9 | 4947.2 | 5693.9 | 5453.7 |
| | 2 | 6260.7 | 4943.8 | 5690.0 | 5450.0 |
| | 3 | 4171.3 | 3293.9 | 3791.1 | 3631.1 |
| Maize | 1 | 2074.0 | 3514.4 | 4831.2 | 7419.0 |
| | 2 | 2077.1 | 3519.7 | 4838.5 | 7430.3 |
| | 3 | 1959.5 | 3320.4 | 4564.5 | 7009.5 |

Note: 1—semi-agricultural half pastoral area; 2—pastoral area; and 3—agricultural area.

**Table 10.** Seed fertilizer input parameters.

| Crop | N in Seeds (kg ha$^{-1}$) | P in Seeds (kg ha$^{-1}$) |
|---|---|---|
| Soybean | 5.3 | 0.14 |
| Wheat | 2.1 | 1.24 |
| Maize | 1.6 | 0.15 |

Crop grain and straw harvest are the principal nutrient outputs of agricultural systems. According to information obtained from the field survey, we used a simple proportion to estimate the amount of nutrient output of the crop harvest in this study; the parameters are listed in Table 11 and the others are listed in Tables 7 and 8. In this study, NH$_3$ volatilization and denitrification are considered. The NH$_3$ volatilization loss rate in fertilizers was 25% and the NO$_2$ volatilization loss rate in fertilizers was 1.05%. The N leaching and runoff were estimated using export coefficients [53]. The leaching output rates were 16.0 kg N ha$^{-1}$ yr$^{-1}$. The runoff output rates were 2.88 kg N ha$^{-1}$ yr$^{-1}$ and 0.23 kg P ha$^{-1}$ yr$^{-1}$.

**Table 11.** Grain fertilizer output parameters.

| Crop | N in Grain (%) | P in Grain (%) |
|---|---|---|
| Soybean | 5.3 | 0.48 |
| Wheat | 2.1 | 0.41 |
| Maize | 1.6 | 0.27 |

## 3. Results

### 3.1. Precision Verification

We used remote sensing data to extract the spatial layout of crops over large areas. The accuracy of extraction depends on the resolution of the remote sensing image and the setting of the NDVI threshold of the decision tree models. The accuracy verification of this study was performed in terms of spatial precision and area accuracy [54]. According to statistics, the area accuracy of the classification was >80%. The accuracy of the training sample extraction model was >82%, the producer's accuracy was >87%, the user's accuracy was >84%, and the kappa was >0.84 (Table 12). This shows that the accuracy was relatively high and meets the application requirements of this study. Additionally, it shows that it is feasible to use phenology information for division crop layout, and that the decision tree classification methods can be applied to the extraction planting patterns.

**Table 12.** Accuracy assessment of crop maps.

| Years | Crop | Statistical Data ($\times 10^4$ ha) | Remote Sensing Data ($\times 10^4$ ha) | Area Precision (%) | Producer's Accuracy (%) | User's Accuracy (%) | Overall Accuracy (%) | Kappa |
|---|---|---|---|---|---|---|---|---|
| 2000 | Maize | 48.3 | 34.5 | 71.4 | 95.1 | 95.0 | 85.6 | 0.87 |
|  | Wheat | 10.7 | 8.0 | 74.9 | 81.2 | 83.3 |  |  |
|  | Soybean | 10.1 | 10.6 | 95.6 | 87.0 | 73.0 |  |  |
| 2005 | Maize | 70.5 | 74.7 | 94.0 | 96.0 | 87.0 | 88.2 | 0.87 |
|  | Wheat | 4.5 | 3.5 | 76.6 | 75.0 | 88.2 |  |  |
|  | Soybean | 8.6 | 7.9 | 91.6 | 95.0 | 90.0 |  |  |
| 2010 | Maize | 104.0 | 103.1 | 99.1 | 90.5 | 75.0 | 82.3 | 0.85 |
|  | Wheat | 3.9 | 2.7 | 70.0 | 86.0 | 82.9 |  |  |
|  | Soybean | 8.1 | 6.6 | 81.1 | 73.1 | 90.0 |  |  |
| 2015 | Maize | 131.2 | 143.6 | 90.5 | 89.5 | 73.9 | 83.6 | 0.78 |
|  | Wheat | 4.4 | 3.1 | 70.5 | 85.3 | 92.9 |  |  |
|  | Soybean | 1.6 | 1.3 | 79.7 | 96.5 | 82.9 |  |  |

*3.2. Description of Crop Planting Spatial Pattern*

3.2.1. Temporal and Spatial Distribution Characteristics of Crops

The crop planting area changed significantly in the West Liaohe River Basin from 2000–2015 (Figure 5). The crop planting area increased from $53.1 \times 10^4$ ha in 2000 to $148.0 \times 10^4$ ha in 2015, i.e., an 180% increase. The planting area of maize increased by 316% from 2000–2015, and the spatial layout of maize was clearly consistent with the distribution of the major tributaries of the West Liaohe River. The soybean planting area showed a decreasing trend, decreasing from $10.6 \times 10^4$ ha in 2000 to $1.3 \times 10^4$ ha in 2015, i.e., an 87.6% decrease, which was mainly concentrated in the Wulijimu River Basin (middle reaches) and West Liaohe River lower reaches basin (below Sujiabao) (lower reaches). The wheat planting area first decreased and then increased, and the overall decreasing trend was reduced from $8.0 \times 10^4$ ha in 2000 to $2.7 \times 10^4$ ha in 2010, and then increased to $3.1 \times 10^4$ ha in 2015, i.e., a decrease of 61.4% which was scattered throughout the study area.

3.2.2. Major Crops Transfer Changes from 2000–2015

The transfer change of the main crops was mainly represented by the conversion of soybean to maize and wheat to maize from 2000–2015 (Figure 6). The soybean area in maize was $4.4 \times 10^4$ ha, and the area of wheat in maize was $3.1 \times 10^4$ ha (Figure 7). The reduction in the soybean planting area was mainly due to the conversion of soybean to maize, and the reduction of wheat planting area was mainly due to conversion of wheat to maize. Although most soybean and wheat was converted to maize, the main reason for the maize planting area increase was other land types (garden, forest, and grassland) into maize, which accounted for 90% of the increased area of maize.

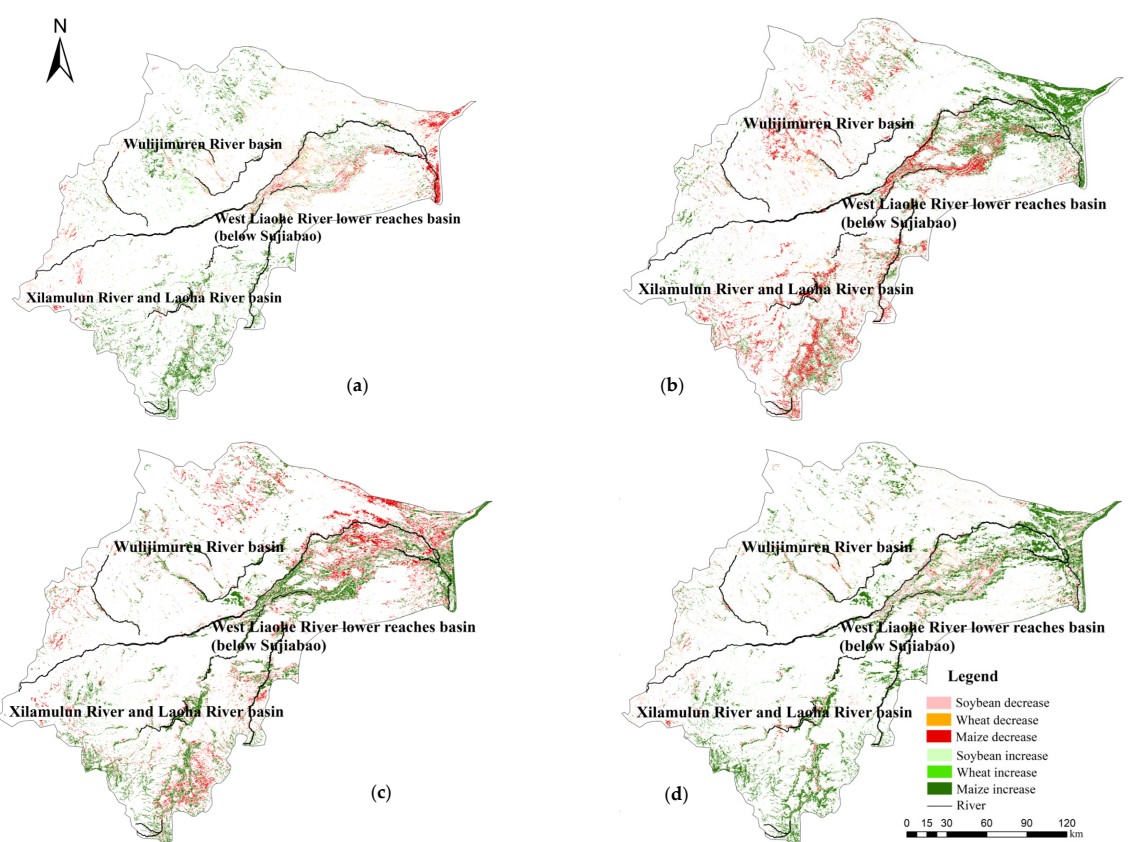

**Figure 5.** Crop layout in (**a**) 2000–2005, (**b**) 2005–2010, (**c**) 2010–2015, and (**d**) 2000–2015.

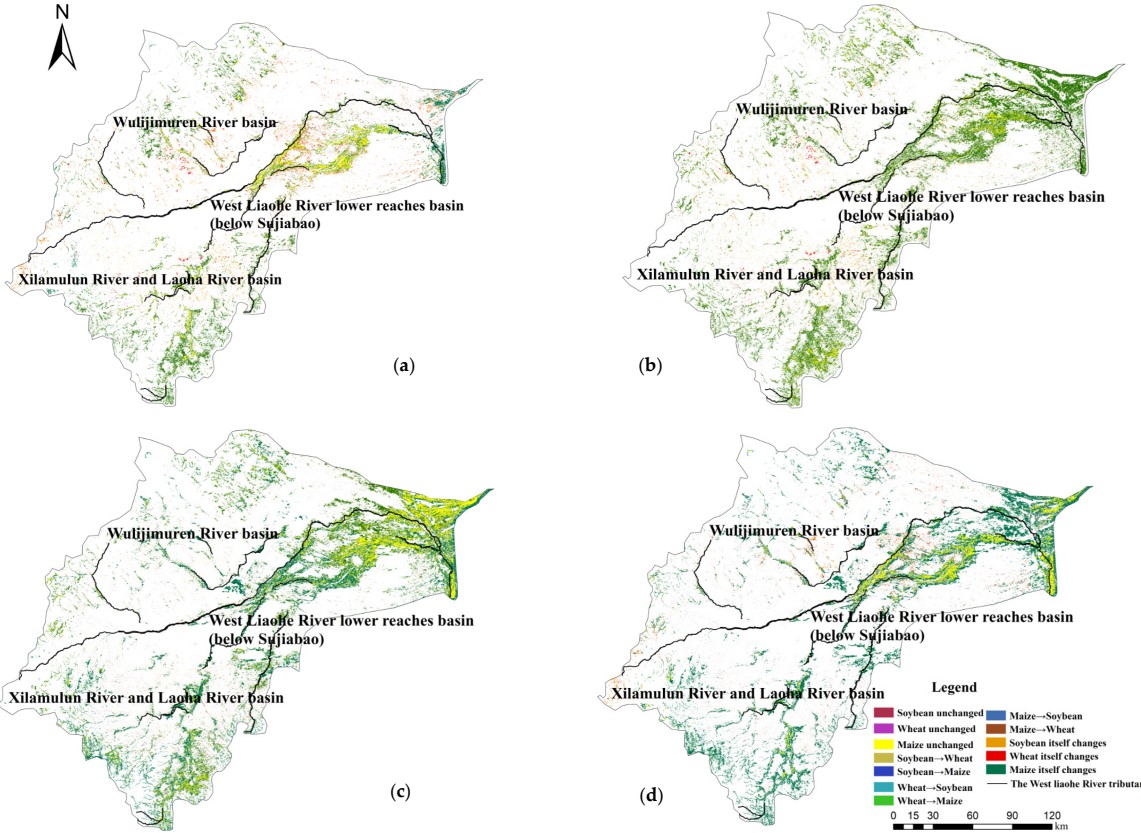

**Figure 6.** Major crop transfer changes in (**a**) 2000–2005, (**b**) 2005–2010, (**c**) 2010–2015, and (**d**) 2000–2015.

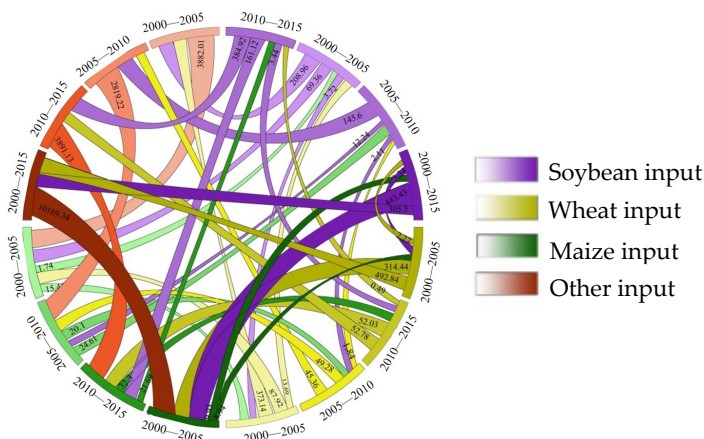

**Figure 7.** Major crop transfer changes in 2000, 2005, 2010, and 2015 ($\times 10^2$ ha).

### 3.3. N and P Input–Output and Balance

#### 3.3.1. N and P Input–Output

During 2000–2015, the total N inputs and outputs increased from $237.7 \times 10^3$ t N and $130.3 \times 10^3$ t N, respectively, to $725.4 \times 10^3$ t N and $142.4 \times 10^3$ t N, respectively (Figure 8a), which is equivalent to mean values of 469.6 and 165.4 kg N ha$^{-1}$ input and of 573.5 and 348.2 kg N ha$^{-1}$ output, respectively. From 2000–2010, the change in total N inputs and outputs showed an upward trend, and the total N input increased more than the N output. Subsequently, the total input continued to increase from 2010–2015, and the N output decreased. The changes in N unit input and output showed a decreasing trend from 2000–2005 and an increase from 2005–2015. The N unit input was greater than the output (Figure 8c). During 2000–2015, total amounts of P input and output increased from $96.3 \times 10^3$ t P and $7.1 \times 10^3$ t P, respectively, to $401.8 \times 10^3$ t P and $44.1 \times 10^3$ t P, respectively (Figure 8b), which is equivalent to mean values of 190.3 and 14.1 kg P ha$^{-1}$ to 317.6 and 34.8 kg P ha$^{-1}$, respectively. The changes in P input and output showed an increasing trend from 2000–2015, and the increase in P input was greater than the increase in P output from 2000–2015. The P unit input and output showed an increasing trend, and the P unit input was greater than the output from 2000–2015 (Figure 8d).

For soybean, the change in total N input from 2000–2015 showed an initial trend of increasing and then decreasing, and the change in total N output showed a downward trend. The change in the total P inputs and outputs showed a trend of initially increasing and then decreasing. For wheat, the change in total N and P inputs and outputs showed a decreasing trend from 2000–2010, and the change in total N and P inputs and outputs showed an upward trend from 2010–2015. For maize, the changes in total N and P inputs and outputs showed an increasing trend from 2000–2015. Unit amounts of N input and output of soybean increased from 670.0 kg N ha$^{-1}$ and 216.7 kg N ha$^{-1}$, respectively, in 2000, to 745.9 kg N ha$^{-1}$ and 238.4 kg N ha$^{-1}$, respectively, in 2015. Unit amounts of N input and output of wheat increased from 1149.8 kg N ha$^{-1}$ and 242.0 kg N ha$^{-1}$, respectively, in 2000, to 1325.8 kg N ha$^{-1}$ and 294.2 kg N ha$^{-1}$, respectively, in 2015. Unit amounts of N input and output of maize increased from 239.1 kg N ha$^{-1}$ and 130.3 kg N ha$^{-1}$, respectively, in 2000, to 552.6 kg N ha$^{-1}$ and 350.7 kg N ha$^{-1}$, respectively, in 2015. The P input of soybean increased from 91.7 kg P ha$^{-1}$ in 2000 to 178.4 kg P ha$^{-1}$ in 2015, while its P output decreased from 15.9 kg P ha$^{-1}$ in 2000 to 15.2 kg P ha$^{-1}$ in 2015. The P input and output of wheat increased from 653.2 kg P ha$^{-1}$ and 28.4 kg P ha$^{-1}$, respectively, in 2000, to 820.0 kg P ha$^{-1}$ and 23.9 kg P ha$^{-1}$, respectively, in 2015. Similarly, the P input and output of maize increased from 108.9 kg P ha$^{-1}$ and 10.0 kg N ha$^{-1}$, respectively, in 2000, to 306.4 kg P ha$^{-1}$ and 35.3 kg P ha$^{-1}$, respectively, in 2015.

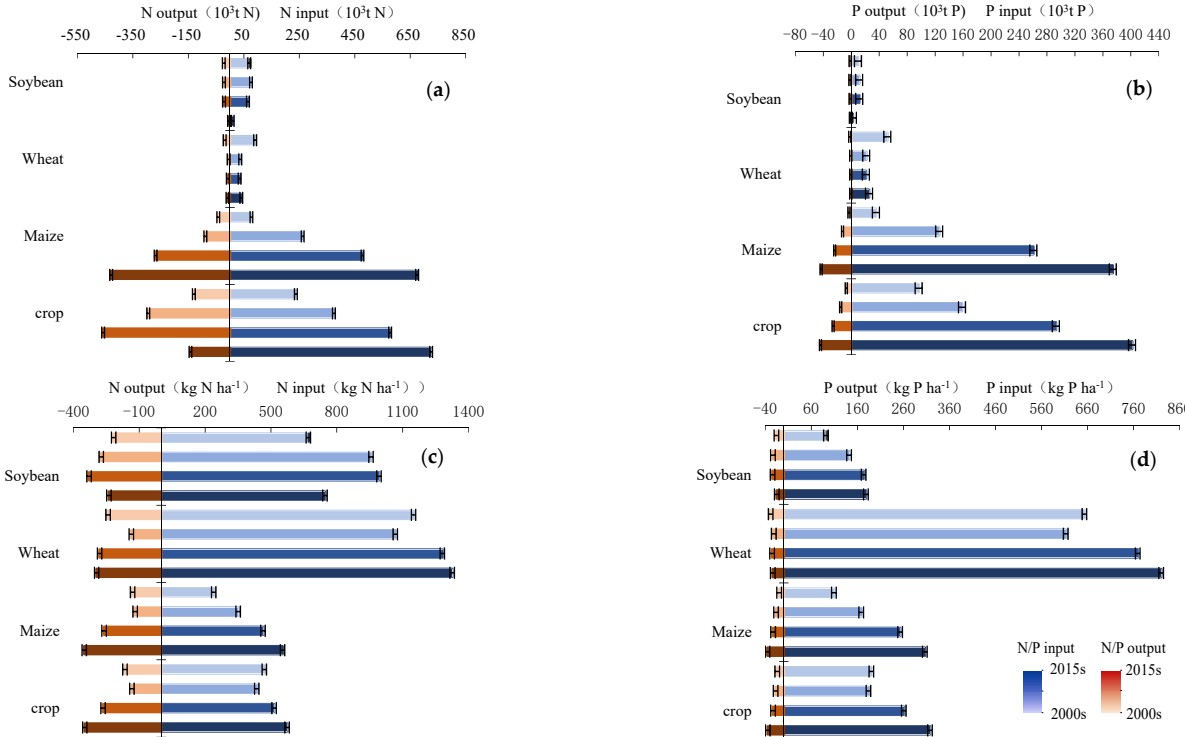

**Figure 8.** (**a**,**c**) N and (**b**,**d**) P nutrient inputs and outputs.

From 2000–2015, the total amounts of N and P inputs were greater than the outputs. The total N inputs and outputs decreased from maize > soybean > wheat. The total amounts of P input and output decreased from maize > wheat > soybean. Unit amounts of N input and output decreased from wheat > soybean > maize. The unit amounts of P input and output decreased from wheat > maize > soybean. It can be seen that the N and P inputs and outputs were not only related to the crop planting area but also to the crop type.

3.3.2. N and P Input–Output Component

To further explore the sources of N and P inputs and outputs, we averaged the ratio of N and P input and output components to the total input and output for 4 years. Chemical fertilizer input and seed input were the main sources of N input, accounting for 49.9% and 28.9%, respectively (Figure 9a). Chemical fertilizer input was the main source of P input, accounting for 64% of the total P input (Figure 9c). Second, organic fertilizer input was also an important source of N and P inputs, accounting for 16.1% and 17.5%, respectively. However, other N and P inputs (such as irrigation, atmospheric deposition, and biological N fixation) had limited effects, and accounted for only 5.1% and 0.5% of total N and P inputs, respectively. Crop harvest output was the main source of N and P output, accounting for 38.7% and 60.2%, respectively. Second, straw harvest was the main source of N and P outputs, accounting for 25.9% and 38.6%, respectively. $NH_3/NO_2$ volatilization, runoff, and leaching were also routes of N and P outputs; if these nutrients are discharged into the atmosphere or water they will cause pollution. It is worth noting that the $NH_3/NO_2$ volatilization accounted for a large proportion of the N output, which was 25.9%. Other N and P outputs (such as leaching and runoff) accounted for a small proportion of N and P outputs, accounting for only 9.5% and 1.2% of total N and P outputs, respectively (Figure 9b,d).

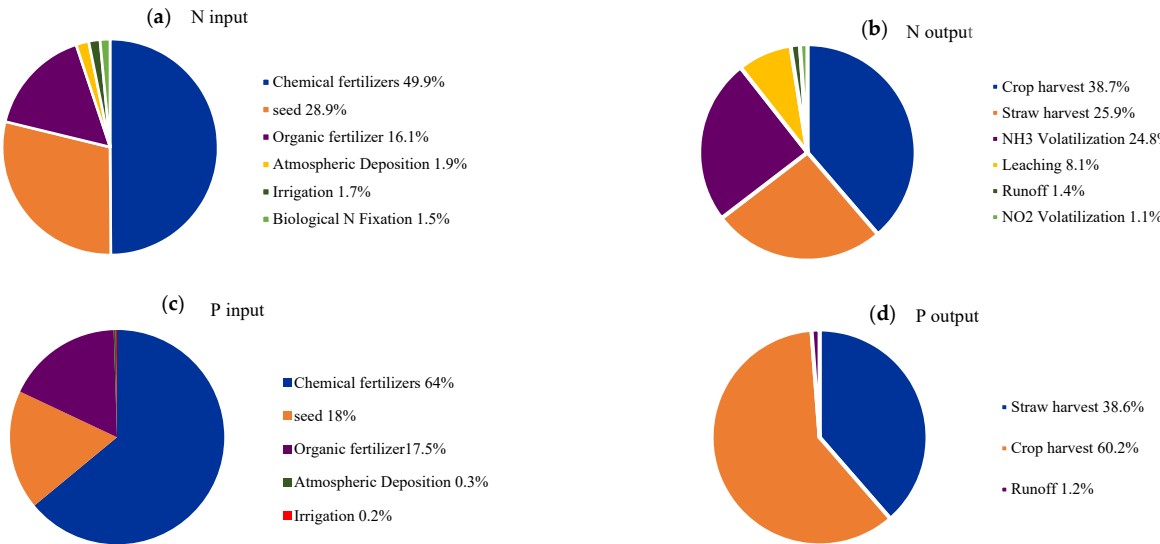

**Figure 9.** Components of (**a**,**b**) N and (**c**,**d**) P inputs and outputs.

### 3.3.3. N and P Balance

The N and P balance exhibited an N and P surplus, and the N and P inputs were greater than the outputs in the West Liaohe River Basin from 2000–2015. The unit N surplus showed a downward trend from 304.2 kg N ha$^{-1}$ in 2000 to 225.3 kg N ha$^{-1}$ in 2015. The N surplus of soybean first increased and then decreased, with an overall increasing trend. The N surplus of soybean increased from 453.4 kg N ha$^{-1}$ in 2000 to 507.5 kg N ha$^{-1}$ in 2015, i.e., an increase of 11.9%, with an mean annual increase of 0.8%. The N surpluses of wheat and maize showed increasing trends. The N surplus of wheat increased from 907.8 kg N ha$^{-1}$ in 2000 to 1031.7 kg N ha$^{-1}$ in 2015, i.e., an increase of 13.6%, with an mean annual increase of 0.9%. The N surplus of maize increased from 108.9 kg N ha$^{-1}$ in 2000 to 201.9 kg N ha$^{-1}$ in 2015 (Figure 10a), i.e., an increase of 85.5%, with a mean annual increase of 5.7%. In terms of crop types, the N surplus decreased from wheat > soybean > maize. The unit P surplus increased overall, from 176.3 kg P ha$^{-1}$ in 2000 to 282.8 kg P ha$^{-1}$ in 2015. The P surpluses of soybean, wheat, and maize showed increasing trends. The P surplus of soybean increased from 75.5 kg P ha$^{-1}$ in 2000 to 163.2 kg P ha$^{-1}$ in 2015, i.e., an increase of 120%, with a mean annual increase of 7.7%. The P surplus of wheat increased from 624.9 kg P ha$^{-1}$ in 2000 to 796.1 kg P ha$^{-1}$ in 2015, i.e., an increase of 27.4%, with an mean annual increase of 1.8%. The P surplus of maize increased from 98.9 kg P ha$^{-1}$ in 2000 to 271.1 kg P ha$^{-1}$ in 2015 (Figure 10b), i.e., an increase of 170%, with an mean annual increase of 11.6%. In terms of crop types, the P surplus decreased from wheat > maize > soybean.

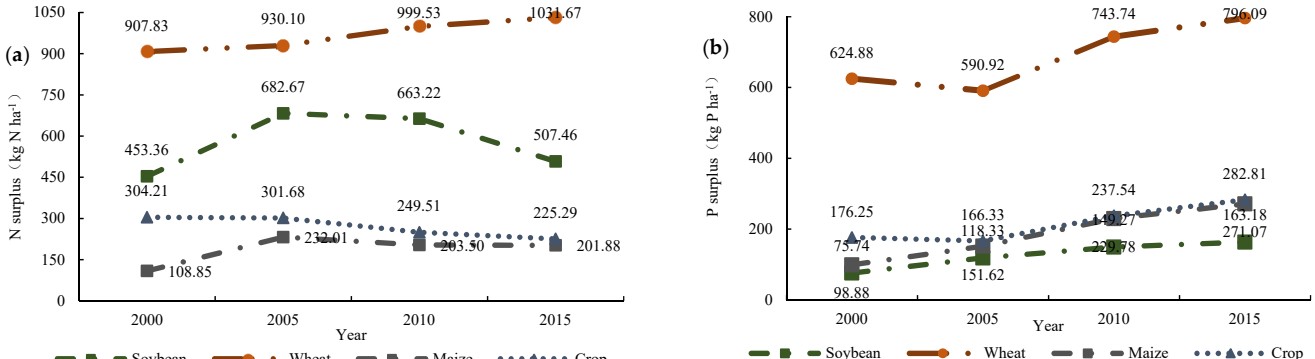

**Figure 10.** (**a**) N and (**b**) P nutrient balance.

### 3.4. N and P Use Efficiencies and Surplus Ratios

N use efficiency and surplus ratio were 20.6 and 64.8% in 2000, 26.3 and 69.3% in 2005, 28.1 and 48.5% in 2010, and 36.3 and 39.3% in 2015, respectively (Figure 11a). P utilization efficiency and surplus ratio were 7.2 and 92.6% in 2000, 9.4 and 90.4% in 2005, 8.8 and 91.1% in 2010, and 10.9 and 89.0% in 2015, respectively (Figure 11b). Spatially, the N surplus ratio changed significantly (Figure 12) and the N surplus changed smoothly (Figure 13). In 2015, the N surplus was concentrated in 35–45% (Figure 12d) and the P surplus was concentrated in 85–90% (Figure 13d) in the West Liaohe River Basin. NUE, NSR, PUE, and PSR values all showed differences between time and crop type. Crop NUE increased to its highest value between 2000 and 2015. Compared with NUE, crop NSR showed an inverse trend from 2005–2015 and decreased to the lowest values in 2015. For different crop types, the soybean NUE increased to its highest value from 2000–2005 and then decreased to the lowest values from 2010–2015. Compared with the soybean NUE, the soybean NSR showed an inverse trend from 2000–2015. The wheat NUE decreased to its lowest values from 2000–2015. Compared with NUE, the wheat NSR increased to its highest value in 2005 and decreased to its lowest value in 2015. It is worth noting that the maize NUE increased to its highest value from 2000–2015. Compared with the maize NUE, the maize NSR showed an inverse trend from 2005–2015 and decreased to the lowest values in 2015. The variation in the NUE and NSR of maize was the same as that of the crop from 2000–2015 (Figure 11a). Soybean and wheat PUE values decreased to their lowest values from 2000–2015. Compared with PUE, the soybean and wheat PSR values showed inverse trends, and the soybean and wheat PUE values increased to the highest value from 2000–2015. It is worth noting that the maize PUE increased to its highest value from 2000–2015. Compared with PUE, the maize PSR showed an inverse trend. The maize PSR decreased to its lowest value in 2015. This was the same as the change in the crop PUE and PSR (Figure 11b).

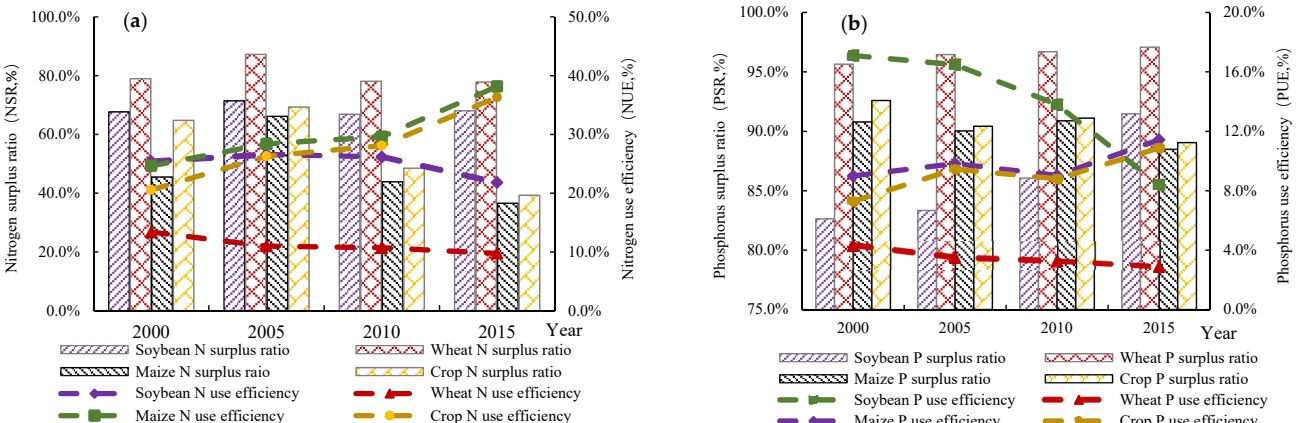

**Figure 11.** (**a**) N and P use efficiency and (**b**) N and P surplus ratios.

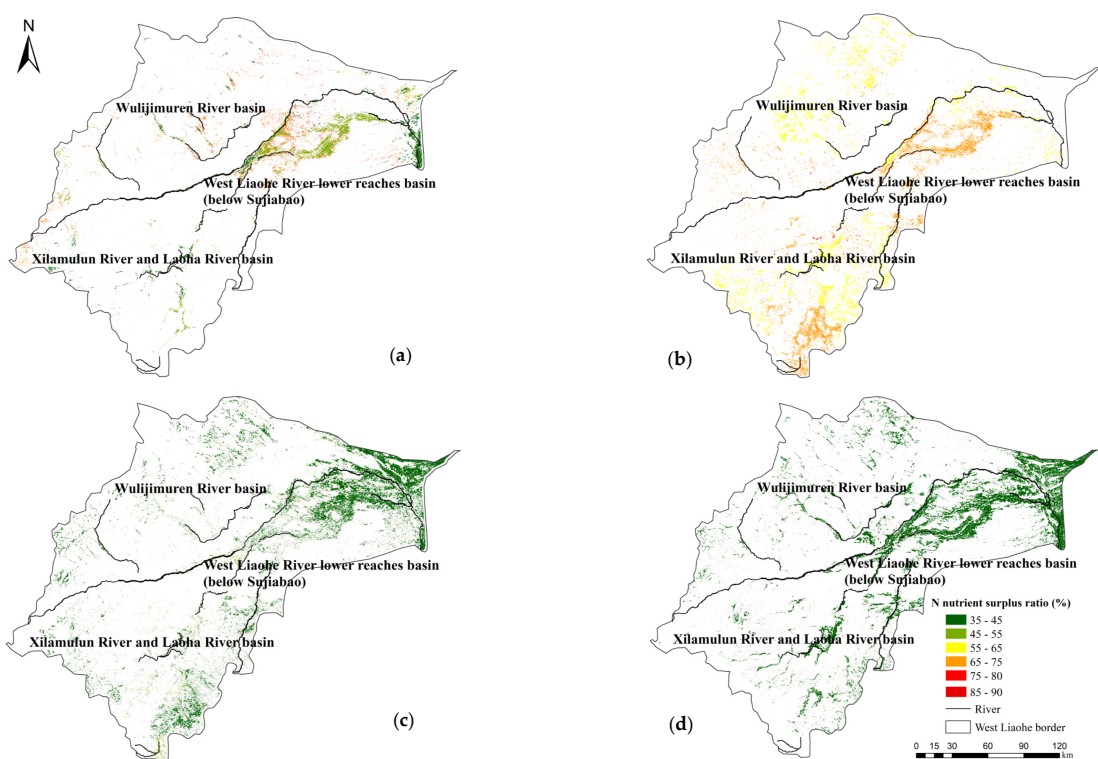

**Figure 12.** Spatial distribution of the N nutrient surplus ratio in (**a**) 2000, (**b**) 2005, (**c**) 2010, and (**d**) 2015.

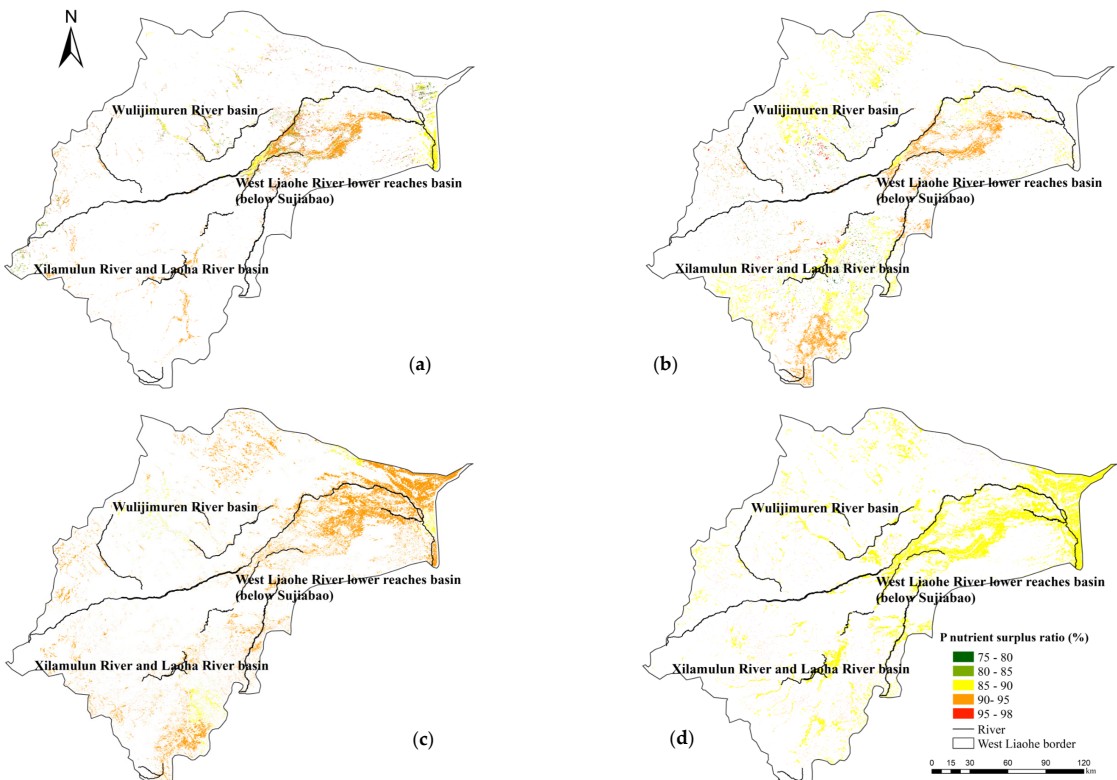

**Figure 13.** Spatial distribution of the P nutrient surplus ratio in (**a**) 2000, (**b**) 2005, (**c**) 2010, and (**d**) 2015.

## 4. Discussion

### 4.1. Crop Layout, Inner Mongolia Agricultural Policy, and Impact on N and P Balance

Changes in maize and soybean planting layouts had a significant impact on N and P nutrient balance. There was a significant positive correlation between the proportion of soybean planting area and the unit N surplus ($R^2$ = 0.0.539, $p < 0.001$) and a significant negative correlation with the unit P surplus ($R^2$ = 0.506, $p < 0.001$). There was a significant negative correlation between the proportion of maize planting area and unit N surplus ($R^2$ = 0.8391, $p < 0.001$) and a significant positive correlation with unit P surplus ($R^2$ = 0.7867, $p < 0.001$). This indicates that the higher the proportion of the soybean planting area, the larger the N surplus and the smaller the P surplus, and the higher the proportion of maize planting area, the smaller the N surplus and the larger the P surplus (Figure 14).

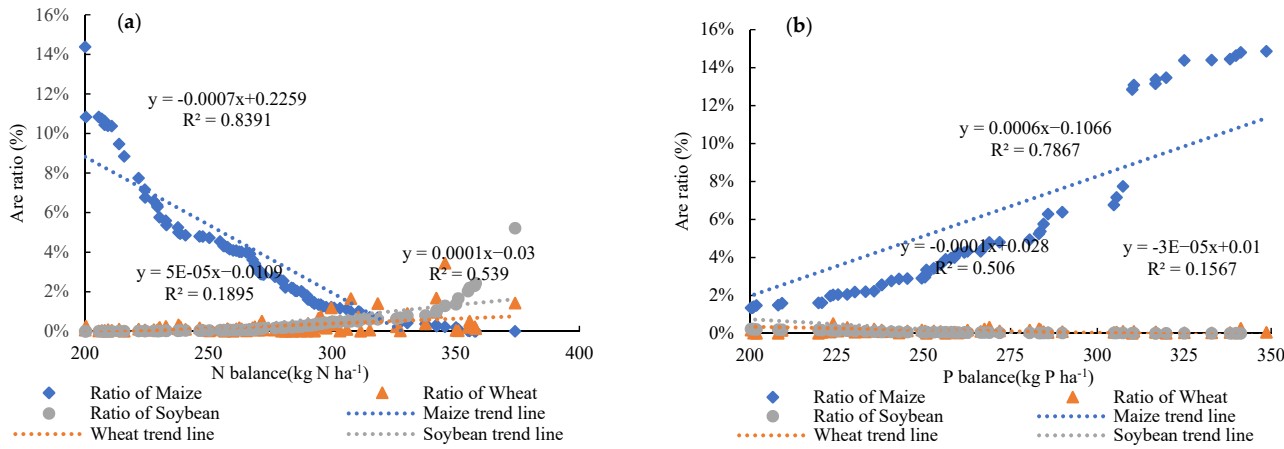

**Figure 14.** Correlations between the crop area ratio and (**a**) N and (**b**) P nutrient balance (by county area).

From 2000–2015, the soybean planting area decreased from $10.6 \times 10^4$ ha in 2000 to $1.3 \times 10^4$ ha in 2015, and the area ratio decreased by 19.04%. The maize planting area increased from $34.5 \times 10^4$ ha in 2000 to $143.6 \times 10^4$ ha in 2015, and the area ratio increased by 32.08%. It can be seen from major crops transfer changes from 2000–2015 that soybean was mainly converted into maize, accounting for 99.8% of the total soybean planting area. To some extent, the conversion of soybean into maize led to a P surplus. However, the planting area of soybean in maize accounted for only 3.97% of the increase in the maize planted area and 93.23% of the new maize planting area was mainly from other land-use types. Therefore, the increase in maize acreage resulted in a decrease in N surplus in the study area. Furthermore, although the ratio of the soybean area to the N and P surplus had a strong correlation, the changes in the maize planting layout played a decisive role in the change in N and P surplus rates and use efficiencies in the West Liaohe River Basin. This is evident from the spatial layout changes of the main crops in the West Liaohe River Basin as well. The layout of the N and P surplus was mainly concentrated in the West Liaohe River lower reaches basin (below Sujiabao) from 2000–2015, and this region also experienced a dramatic change in the maize layout (Figure 15).

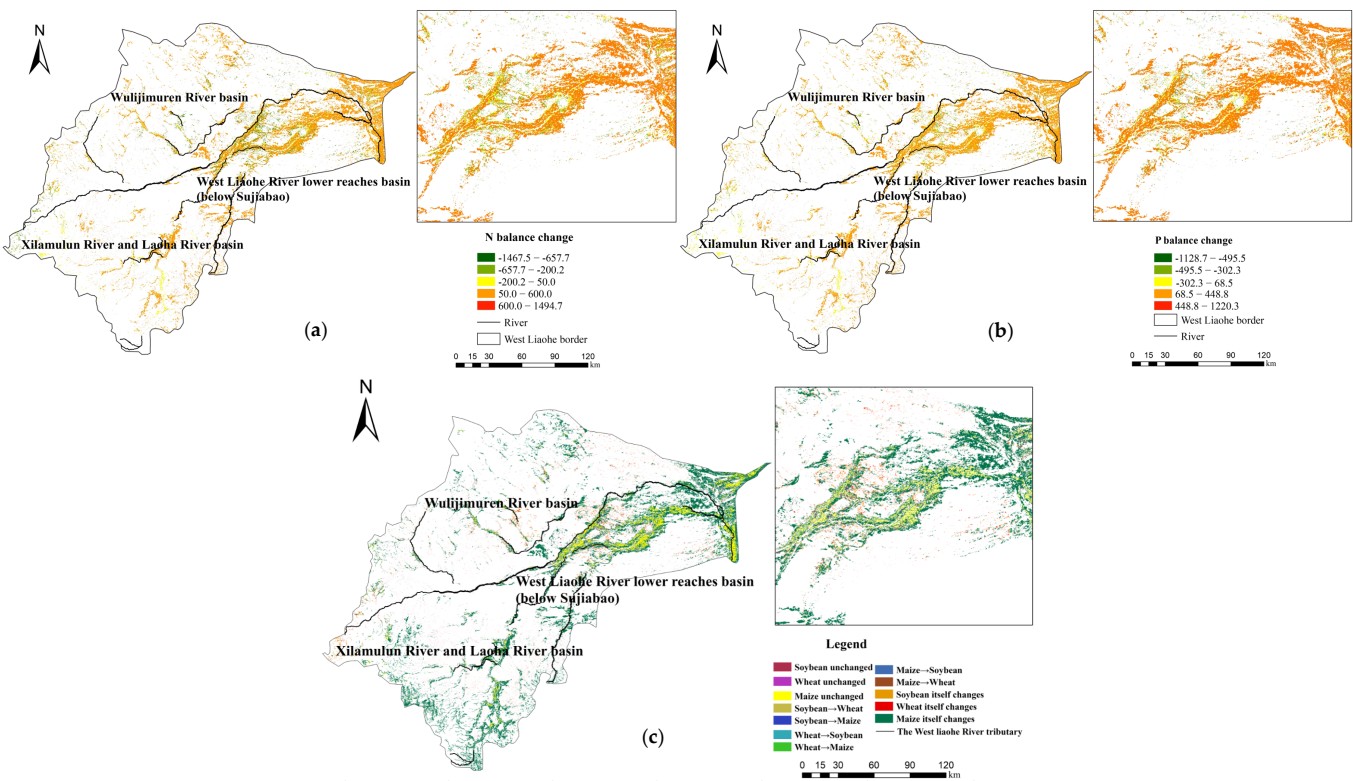

**Figure 15.** (**a**,**b**) N and P balance and (**c**) crop layout change.

### 4.2. N and P Surplus and Non-Point Source Pollution

In the West Liaohe River Basin, the N and P residuals per unit area were 225.29–304.21 kg N ha$^{-1}$ and 176.25–282.81 kg P ha$^{-1}$, respectively. These concentrations far exceed the residual risk threshold of 180 kg N ha$^{-1}$ [55] and 35 kg P ha$^{-1}$ [56], respectively; hence, there is a risk of environmental pollution. Chemical fertilizer is the main source of N and P nutrient inputs, and the amount of fertilizer per unit area of farmland in the West Liaohe River Basin was up to 417.93 kg ha$^{-1}$, which considerably exceeds the safety limit of 225 kg ha$^{-1}$ set by developed countries to prevent chemical fertilizers from harming the environment [57]. From the N and P input components, fertilizer N and P nutrient inputs were three and four times higher than the organic fertilizer N and P nutrient inputs, respectively. Therefore, reducing N and P surplus and applications of chemical fertilizer N and P are important. The West Liaohe River Basin is an alternative farming and pastoral zone, and the combination of farming and pastoral farming is conducive to improving N use efficiency. Manure produced by livestock and poultry breeding in the West Liaohe River Basin can be absorbed by farmland, and we can reasonably set the ratio of organic fertilizer and chemical fertilizer, reduce the amount of chemical fertilizer application, and replace chemical fertilizer with organic fertilizer.

### 4.3. Future Potential Steps and Research Needs

By consulting the relevant literature it is evident that there has been less research on the farmland nutrient balance in Inner Mongolia, and that the majority of recent studies have been soil experimental studies in small areas. Such research cannot highlight the differences between regions or guide large-area control of agricultural non-point source pollution. In this study, we established a GIS database of N and P balance based on long-term serial MOSDIS data at the pixel scale. The results provide guidance for the adjustment of regional planting structures and the prevention and control of agricultural non-point source pollution. In the future, it will be necessary to continuously optimize the remote sensing interpretation model, improve the accuracy of crop interpretation, and identify the optimal ratio of organic and chemical fertilizer inputs to optimize crop planting layouts.

## 5. Conclusions

In this study, we evaluated the impact of crop layout changes on N and P nutrient balance in the West Liaohe River Basin from 2000–2015. We found that the crop planting area showed an increasing trend, and that the spatial layout was consistent with the major tributaries of the West Liaohe River Basin. The spatial layout of maize was highly variable. Most soybean and wheat cropping areas had been converted into maize; however, the main reason for the increase in the maize planting area was the conversion of other land types (e.g., gardens, woodlands, and grasslands) to maize. The total N and P inputs and outputs showed increasing trends from 2000–2015. N and P inputs and outputs were not only related to the crop planting area but also to the crop type. Chemical fertilizer input was the main source of N and P inputs. We also found that the study area has been in a state of N and P surplus for a long time period. In the West Liaohe River Basin, the N and P residuals per unit area were 225.29–304.21 kg N ha$^{-1}$ and 176.25–282.81 kg P ha$^{-1}$, respectively. These values far exceeded the residual risk threshold of 180 kg N ha$^{-1}$ and 35 kg P ha$^{-1}$, respectively, and the N and P use efficiency was low; therefore, there was a risk of agricultural non-point source pollution.

Further research found that changes in the spatial and temporal layout of crops have an important influence on the N and P nutrient balance in the study area. There was a significant positive correlation between the proportion of the soybean planting area and the unit N surplus, and a significant negative correlation with the unit P surplus. There was a significant negative correlation between the proportion of the maize planting area and unit N surplus, and a significant positive correlation with the unit P surplus. The layout of the N and P surpluses were mainly concentrated in the West Liaohe River lower reaches basin (below Sujiabao) from 2000–2015, and this region has also experienced dramatic changes in the maize layout. Therefore, the changes of maize planting layout has played a decisive role in the change of N and P surplus rates and use efficiencies. It is necessary to properly control the maize and soybean planting areas, and appropriately reduce the wheat planting area in the study area. In addition, the West Liaohe River Basin is an alternate farming and pastoral zone, and the combination of farming and pastoral is conducive to the improvement of N and P use efficiencies. Manure produced by livestock and poultry breeding can be absorbed by farmland, and we can reasonably set the ratio of organic fertilizer and chemical fertilizer and reduce the amount of chemical fertilizer application.

**Author Contributions:** Conceptualization, D.L.; data curation, Z.Z.; formal analysis, D.L. and Z.Z.; funding acquisition, D.L.; investigation, D.L. and Z.Z.; methodology, D.L. and Z.Z.; project administration, D.L.; resources, D.L.; software, B.F. and Z.Z.; supervision, D.L.; validation, D.L. and Z.Z.; visualization, Z.Z. and B.F.; writing—original draft, Z.Z.; writing—review and editing, D.L., Z.Z. and B.F. All authors have read and agreed to the published version of the manuscript.

**Funding:** This research was supported by the National Natural Science Foundation of China (Grant No. 41671525), the National Key Research and Development Program of China (Grant No. 2016YFC0503500), and Basic Research Fund of Chinese Academy of Agricultural Sciences, China (Y2020YJ18).

**Institutional Review Board Statement:** Not applicable.

**Informed Consent Statement:** Informed consent was obtained from all subjects involved in the study.

**Data Availability Statement:** Data sharing not applicable.

**Acknowledgments:** The authors gratefully acknowledge National Natural Science Foundation of China, the Ministry of Science and Technology of China, and the Chinese Academy of Agricultural Sciences, China, for financial support.

**Conflicts of Interest:** The authors declare no conflict of interest.

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
