# Peer review of "Evaluating the Impact of Crop Layout Changes on N and P Nutrient Balance: A Case Study in the West Liaohe River Basin, China"

_sustainability, doi:10.3390/su13147982_

Round 1
Reviewer 1 Report
Dear Authors,
Please find enclosed some recommendations. In yellow, you will find some punctuations errors (uppercase instead of lowercase, spaces, etc.). On the other hand, I'm not English native, but I have found some incomplete sentences (also in yellow).
Concerning the text, I would recommend you to include some paragraphs as Tables, or even to delete some paragraphs because they are well understood in the Figures (you will find the indication as note inside the text).
The findings are well explained.
Greetings,
The Reviewer

Author Response
Thanks for your suggestion. We are so sorry for our punctuations and sentences errors. We have reorganized the language and logical structure of the article to make the expression clearer. Your suggestions are very helpful to us,we have deleted some paragraphs which are already reflected in the figure. We hope you are satisfied with our response.
Point 1: In yellow,you will find some punctuations errors (uppercase instead of lowercase, spaces, etc.). 

Response 1: Thanks for your suggestion. We are so sorry for our punctuations and sentences errors. We have reorganized the language and logical structure of the article to make the expression clearer.
Point 2: On the other hand,I'm not English native,but I have found some incomplete sentences (also in yellow).
Response 2: Thank you for your careful review. We are very sorry for the mistakes in this manuscript and inconvenience they caused in your reading. The manuscript has been thoroughly revised and rewritten by a native English speaker, so we hope it can meet the journal’s standard.
Point 3: Concerning the text,I would recommend you to include some paragraphs as Tables,or even to delete some paragraphs because they are well understood in the Figures (you will find the indication as note inside the text). 

Response 3: Your suggestions are very helpful to us,we have deleted some paragraphs which are already reflected in the figure. We hope you are satisfied with our response.
We are very grateful to your comments for the manuscript. According to your advice, we amended the relevant part in manuscript. All the questions are answered one by one in the attachment.

Reviewer 2 Report
The manuscript is rather confused in some parts. Language usage should be improved extensively in some specific points of the article. Even the first part of the title should be revised for clarity and consequentiality. The article lacks extensive literature review coving worldwide issues, and not only country or continental aspects of the issue. Pros&cons of the approach are not clarified. It seems to have a customary approach for a case study. What is the novelty of the approach? What is the real contribution to agricultural science and sustainability science?
Author Response
Thanks for your suggestion. We are so sorry for our Language usage and Logic confusion. We have reorganized the language and logical structure of the article to make the expression clearer. Your suggestions are very helpful to us.We have added an extensive literature review of worldwide issues to the introduction and have pointed out innovations in the research methodology both in the introduction and in the methods. We hope you are satisfied with our response.

Reviewer 3 Report
Changes in Crop Layout, Effects on Farmland Nutrient Balance: A Case Study in The West Liaohe River Basin, China
The first part of the title seems confusing. It needs to be revised while the second part is ok. As in title and in the abstract too, it is not clear as to for which the effect on nutrient balance is being diagnosed. But as it transpires later, I think authors are evaluating the impact of crop layout changes on nutrient balance. If it is so, it needs to be clarified in the whole paper starting from title. However, there is limited background on the impact of the former on the later in introductory part. The methods are amply discussed but the section 2.2.4 needs revisit as there are some structural error eg. ‘seperately….’. As well, it is not clear the purpose of nutrient data collection for only 2 years. The results are enriched with important insights however, there is very little relevance given to the previous work in the field. The discussion too lacks significantly reference or validation/contrast of results with the previous work. Hardly a single study is cited in these two sections. The conclusions are slim and need to give some valuable insights for future research, policy and/or limitation of the current work.
Author Response
Thanks for your suggestion. We are so sorry for our Language usage and Logic confusion. We have reorganized the language and logical structure of the article to make the expression clearer. Your suggestions are very helpful to us. In addition,we have revised the title of the article and enriched the conclusion of the paper. We hope you are satisfied with our response.

Round 2
Reviewer 2 Report
Good revision overall. Acceptable pending in-house language revisions.
Author Response
Dear Reviewer:
The manuscript has been thoroughly revised and rewritten by a native English speaker, so we hope it can meet the journal’s standard.
This study provides new information regarding the impact of crop layout changes on the nitrogen and phosphorus nutrient balance in the West Liaohe River Basin from 2000‒2015. To achieve this, we combined the advantages of remote sensing to interpret crop layout with MODIS data, constructed a farmland nutrient balance model, and constructed a nutrient balance database based on GIS. Among the findings, the study showed significant correlations between planting areas and nutrient surpluses, and that the unit N surplus exhibited a downward trend and the unit P surplus showed an increasing trend. Importantly, the results show that wheat planting area must be reduced and the areas of maize and soybean must be controlled to adjust the N and P balance and mitigate environmental pollution risk. We believe that our study makes a significant contribution to the literature because it demonstrates that combining farming and pastoral farming is conducive to improving N and P use efficiency, and provides guidance for future agricultural practices in the study area.
Further, we believe that this paper will be of interest to the readership of your journal because it is relevant to food security, sustainable crop production, and sustainable utilization of resources such as land, water, atmosphere and other biological resources.
This manuscript has not been published or presented elsewhere in part or in entirety and is not under consideration by another journal. We have read and understood your journal’s policies, and we believe that neither the manuscript nor the study violates any of these. Details about competing interests are provided in the manuscript.
Thank you for your consideration.
Kind regards.

Round 3
Reviewer 2 Report
Good revision, thank you. I see some minor points as below. Editors can check without sending it out for review again.
Although I see big improvements, please check the language usage (too technical in some parts, too long sentences at the end of the paper), and also pay attention to:
- literature review, it should be more extensive and international,
- aims and scope, not completely clear at now,
- discussion/conclusion, future studies should be clarified.
Author Response
Thanks for your suggestion. The manuscript has been thoroughly revised and rewritten by a native English speaker, so we hope it can meet the journal’s standard.
